# Snail augments fatty acid oxidation by suppression of mitochondrial ACC2 during cancer progression

Ji Hye Yang[1],*, Nam Hee Kim[1],* , Jun Seop Yun[1], Eunae Sandra Cho[1] , Yong Hoon Cha[2], Sue Bean Cho[1], Seon-Hyeong Lee[3], So Young Cha[1], Soo-Youl Kim[3], Jiwon Choi[1], Tin-Tin Manh Nguyen[4] , Sunghyouk Park[4], Hyun Sil Kim[1] , Jong In Yook[1]

Despite the importance of mitochondrial fatty acid oxidation (FAO) in cancer metabolism, the biological mechanisms responsible for the FAO in cancer and therapeutic intervention based on catabolic metabolism are not well defined. In this study, we observe that Snail (SNAI1), a key transcriptional repressor of epithelial–mesenchymal transition, enhances catabolic FAO, allowing pro-survival of breast cancer cells in a starved environment. Mechanistically, Snail suppresses mitochondrial ACC2 (ACACB) by binding to a series of E-boxes located in its proximal promoter, resulting in decreased malonyl-CoA level. Malonyl-CoA being a well-known endogenous inhibitor of fatty acid transporter carnitine palmitoyltransferase 1 (CPT1), the suppression of ACC2 by Snail activates CPT1-dependent FAO, generating ATP and decreasing NADPH consumption. Importantly, combinatorial pharmacologic inhibition of pentose phosphate pathway and FAO with clinically available drugs efficiently reverts Snail-mediated metabolic reprogramming and suppresses in vivo metastatic progression of breast cancer cells. Our observations provide not only a mechanistic link between epithelial–mesenchymal transition and catabolic rewiring but also a novel catabolism-based therapeutic approach for inhibition of cancer progression.

## Introduction

During the natural history of human solid cancer, cancer cells repeatedly encounter a metabolic-starved microenvironment which has to be overcome for successful cancer progression (Aktipis et al, 2013). Although glucose is largely regarded as a major source of anabolic cancer cell metabolism, aerobic glycolysis is inefficient in providing adenosine 5'-triphosphate (ATP) (Vander Heiden et al, 2009). Interestingly, quantitative metabolomics analysis from clinical samples have revealed that solid cancer tissue exhibits extremely low glucose levels due to the limited distance of glucose diffusion from functional tumor blood vessels (Walenta et al, 2003; Hirayama et al, 2009). Nonetheless, ATP levels in the clinical samples were well maintained in the glucose-starved tumor microenvironment (Walenta et al, 2003; Hirayama et al, 2009), suggesting that essential ATP may be generated from something other than glucose. During metastatic cancer progression, matrix-detached cancer cells also encounter ATP deficiency and oxidative stress due to loss of glucose transport (Schafer et al, 2009). In these starved conditions, therefore, ATP, mainly from oxidative phosphorylation, as well as NADPH for reductive biosynthesis, are essential metabolites required for overcoming metabolic stress and for successful cancer progression, although catabolic reprogramming by oncogenic signaling is not fully understood.

Fatty acid metabolism consists of the anabolic process of fatty acid synthesis (FAS) under nourished condition and the catabolic process of fatty acid oxidation (FAO) in starved environment (Foster, 2012). The mutually exclusive FAS and FAO are reciprocally dependent on nutritional status, acetyl-coenzyme A carboxylases (ACCs) playing crucial roles in such reciprocal fatty acid metabolism (Foster, 2012; Jeon et al, 2012). In particular, mitochondrial ACC2 determines the switch between FAS and FAO by catalyzing the carboxylation of acetyl-CoA to produce malonyl-CoA, a potent endogenous inhibitor of carnitine palmitoyltransferase 1 (CPT1) (Qu et al, 2016). Because CPT1 is a rate-limiting enzyme of FAO responsible for acyl-carnitine transport into the mitochondria, ACC2 (acetyl-coA carboxylase beta, ACACB) activity and abundance are tightly controlled in many tissues, including cancer cells. The AMPK (5' AMP-activated protein kinase) is a well-known regulator which suppresses ACC enzymatic activity, resulting in ATP and NADPH homeostasis (Jeon et al, 2012). Although the importance of FAO in metastatic progression in human cancer has recently been reported (Lee et al, 2019), the upstream regulators and their functional relevance in cancer progression are not fully understood.

[1]Department of Oral Pathology, Oral Cancer Research Institute, Yonsei University College of Dentistry, Seoul, Korea   [2]Department of Oral and Maxillofacial Surgery, Yonsei University College of Dentistry, Seoul, Korea   [3]Tumor Microenvironment Research Branch, National Cancer Center, Ilsan, Korea   [4]Natural Product Research Institute, College of Pharmacy, Seoul National University, Seoul, Korea

Correspondence: jiyook@yuhs.ac; khs@yuhs.ac
*Ji Hye Yang and Nam Hee Kim contributed equally to this work

Snail is a transcriptional repressor whose aberrant expression has been closely linked to cancer cell epithelial–mesenchymal transition (EMT) and cancer progression (Cano et al, 2000). Major oncogenic pathways, such as Wnt oncogene and p53 tumor suppressor, modulate Snail activities (Yook et al, 2006; Kim et al, 2011), suggesting that transcriptional repression by Snail plays a key role during cancer progression. Whereas earlier studies have reinforced phenotypic conversion and migratory potential during EMT, recent evidence indicates that EMT of cancer cells is also involved in metabolic reprogramming of cancer cells as well as in therapeutic resistance and cancer cell stemness (Vega et al, 2004; Kim et al, 2017). Recently, we have reported that Snail suppresses glycolytic activity via suppression of PFK-1 in cancer cells, resulting in glucose reflux toward the pentose phosphate pathway (PPP) and NADPH generation (Kim et al, 2017). The role of Snail in promoting cancer cell survival under metabolic starvation is evident; the mechanism by which Snail contributes to catabolic ATP generation under starved condition remains unclear. In this study, we found that ACC2 transcript abundance was globally suppressed in many types of human cancer samples compared with adjacent normal tissue. Snail augments FAO, providing essential ATP via transcriptional suppression of mitochondrial ACC2 followed by increased mitochondrial CPT1 activity. Interestingly, pharmacological combinatorial inhibition of PPP and FAO with clinically available drugs successfully interrupts Snail-mediated metabolic reprogramming and metastatic progression in vivo. Our observations provide not only the mechanistic link between Snail-EMT program and catabolic rewiring of cancer cells but also a pharmacologic strategy for breast cancer using metabolic drugs.

# Results

### Snail regulates ATP level via FAO under glucose-starved condition

Snail potentiates cancer cell survival under metabolic stress by activation of PPP and subsequent NADPH generation (Kim et al, 2017). Consistently, knockdown of Snail significantly decreased clonogenic potential in glucose-starved breast cancer cells, whereas overexpression of Snail had the opposite effect (Figs 1A and S1A). Because ATP together with NADPH is essential to cancer cell survival under starved condition, we next measured ATP levels in breast cancer cells after knockdown of Snail. Interestingly, loss of Snail significantly decreased ATP levels, particularly in glucose-starved condition compared with nourished culture condition (Figs 1B and S1B), and inducible expression of Snail significantly augmented ATP levels without glucose (Fig S1C). These results suggest that Snail plays an important role in ATP generation under starved condition in breast cancer cells.

Although ATP can originate from many sources in the metabolic circuit, mitochondrial FAO is the most efficient process for generating ATP, especially in glucose-starved condition, and CPT1 is the gatekeeper of the FAO. To obtain direct evidence that the ATP is generated from fatty acid rather than glucose, we administered palmitate–BSA conjugate as a mitochondrial FAO substrate to breast cancer cells in a glucose-starved environment. Interestingly, palmitate supplement was sufficient to recover ATP levels without

glucose and a CPT1 inhibitor etomoxir (ETX) largely blocked palmitate-derived ATP recovery (Fig 1C), indicating that mitochondrial FAO plays an important role in ATP homeostasis under starved condition. Given the effect of Snail on ATP level, we next analyzed FAO via the incorporation of $^{13}C16$-palmitate to tricarboxylic acid (TCA) cycle intermediates according to the Snail abundance (Veglia et al, 2019). Indeed, incorporation of carbons from FAO to downstream metabolites such as aspartate, malate, and fumarate decreased in Snail knockdown cells as shown by the levels of their M+2 isotopomers (Fig 1D). These results support that Snail plays an important role in ATP homeostasis via FAO in a starved microenvironment.

During mitochondrial FAO, acyl-CoA dehydrogenases (ACADVL and ACADM) and hydroxyacyl-CoA dehydrogenase (HADHA) catalyze the serial steps of the fatty acid $\beta$-oxidation pathway. Examining those enzymatic activities under glucose-starved condition, we found that loss of Snail significantly suppressed FAO enzymatic activities, whereas its overexpression increased those activities in breast cancer cells (Figs 1E and S1D), supporting that Snail abundance is closely correlated with mitochondrial FAO activity.

For FAO in the mitochondrial matrix, the long-chain acyl-CoA is converted to acyl carnitine by CPT1 as a rate-limiting step in FAO on the outer mitochondrial membrane, the acyl carnitine then being transported across the mitochondrial membrane (Qu et al, 2016). Given the role of Snail in FAO, we next explored the CPT1 activity based on Snail abundance in breast cancer cells. Indeed, Snail abundance due to shRNA or overexpression was tightly correlated with CPT1 activity in breast cancer cells (Figs 1F and S1E). The ETX is an irreversible inhibitor of CPT1 and is a clinically available drug for the treatment of type II diabetes and heart failure (Kruszynska & Sherratt, 1987). Regardless of multiple off-target effects, ETX effectively inhibits CPT1 activity as well as depletes intracellular CoA level (Divakiaruni et al, 2018). When we administered ETX in conjunction with induction of Snail, the ETX largely rescued ATP and NADPH levels in breast cancer cells while increasing cell death under glucose-starved condition (Fig 1G). The ETX also suppressed oxygen consumption in breast cancer cells (Figs 1H and S1F). Although CPT1 activity can be regulated at various levels, malonyl-CoA catalyzed by ACC2 is a well-known potent inhibitor of CPT1, constituting a reciprocal control between FAO and FAS (McGarry et al, 1978; Foster, 2012). When we examined malonyl-CoA level quantitatively based on Snail abundance, we found that knockdown of Snail increased malonyl-CoA levels, whereas overexpression of Snail decreased the levels (Figs 1I and S1G). Given the key role of malonyl-CoA in reciprocal switching between FAS and FAO, Snail suppressed FAS in the 3T3-L1 system (Fig S1H). Taken together, these findings indicate that Snail augments FAO by suppressing malonyl-CoA abundance and subsequently relieving the malonyl-CoA–mediated inhibition of CPT1. To explore the mechanistic link between Snail and FAO, we next hypothesized that Snail suppresses malonyl-CoA production and then increases FAO by suppression of ACC (Fig 1J).

### ACC2 is a target of Snail repressor in human cancer

In human and other mammals, two ACC isoforms exist, namely, ACC1 (ACACA) and ACC2 (ACACB). The ACC1 is mainly expressed in lipogenic tissues and catalyzes the committed step in the biosynthesis of long-chain fatty acid. In contrast, ACC2 is localized at the

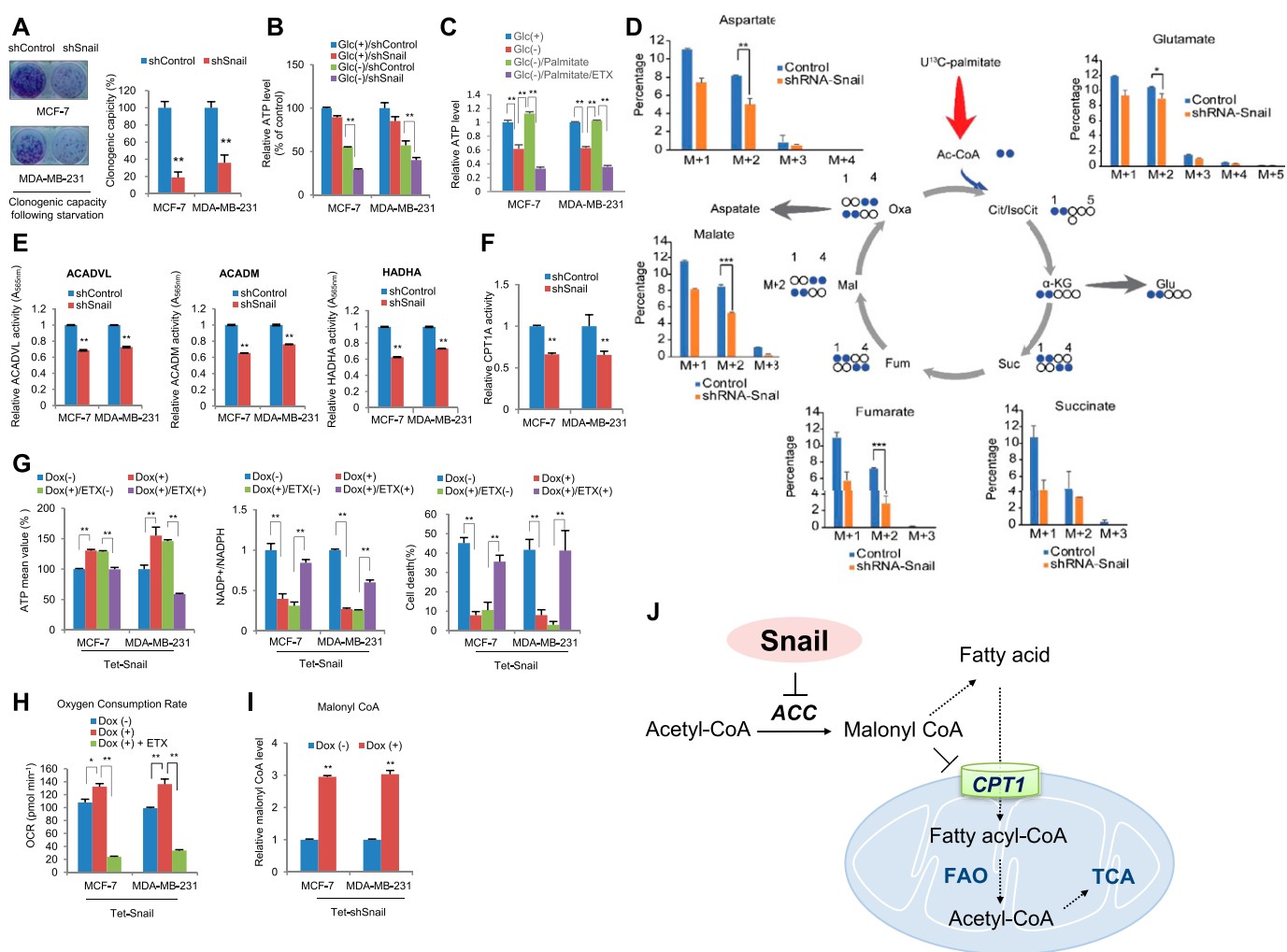

**Figure 1. Snail augments ATP levels via fatty acid metabolism promoting cancer cell survival under glucose starvation.**
**(A)** Clonogenic survival assay of cancer cells following glucose starvation as described in the Materials and Methods section (left). Colonies of more than 50 cells were counted after crystal violet staining (right). Data are expressed as means and SD. The double asterisks denote $P < 0.01$, one asterisk denoting $P < 0.05$ (n = 5, means ± SD, $t$ tests). **(B)** The cancer cells expressing control-shRNA or Snail-shRNA were incubated in the presence (Glc+) or absence (Glc−) of glucose for 4 h, and the relative ATP levels were measured (n = 3, means ± SD, $t$ tests). **(C)** The breast cancer cells in the absence of glucose (Glc−) were treated with BSA-Palmitate (100 $\mu$M) in combination with DMSO control or ETX (100 $\mu$M) for 4 h, and the relative ATP levels were measured (n = 3, means ± SD, $t$ tests). **(D)** The MDA-MB-231 cells expressing control-shRNA or Snail-shRNA were treated with 13C-palmitate (100 $\mu$M) in the absence of glucose for 4 h. Mass isotopomer distribution of [U-13C]-palmitate–derived carbon into some TCA metabolites was determined by LC–MS. Filled blue circles represent 13C atoms derived from [U-13C]-palmitate (*$P < 0.05$, **$P < 0.01$, ***$P < 0.001$, $t$ tests). **(E)** The enzymatic activities of ACADVL, ACADM, and HADHA in breast cancer cells expressing shRNA for control (shControl) or Snail (shSnail) were measured under glucose-starved condition (0.5 mM glucose, n = 3, means ± SD, $t$ tests). **(F)** The CPT1 activities of breast cancer cells expressing shRNA for control (shControl) or Snail (shSnail) under glucose-starved condition were measured (0.5 mM glucose, n = 3, means ± SD, $t$ tests). **(G)** Snail was induced by treatment of doxycycline (Dox) for 48 h and ATP (left), NADPH (middle), and cell death (right) in starved condition were measured in combination with CPT1 inhibitor etomoxir (ETX, 100 $\mu$M, n = 3, means ± SD, $t$ tests). **(H)** Mitochondrial oxygen consumption rate (OCR) in Tet-inducible Snail in combination with ETX (n = 3, means ± SD, $t$ tests). **(I)** Malonyl-CoA abundances in breast cancer cells expressing shRNA for control (shControl) or Snail (shSnail) were measured (n = 3, means ± SD, $t$ tests). **(J)** A schematic diagram depicting a potential mechanism by which the Snail regulates fatty acid oxidation (FAO).

mitochondrial outer membrane and regulates FAO with its malonyl-CoA products (Jeon et al, 2012). Despite the importance of cancer cell metabolism, differential expression of ACC isoforms in human cancer is not clearly understood. Therefore, we next examined differential transcript abundance of ACC1 and ACC2 in clinical cancer samples. For differential expression analysis, we have collected public data files on The Cancer Genome Atlas (TCGA) from many types of human cancer, including normal control samples. Interestingly, mitochondrial ACC2 transcript abundance

was significantly suppressed in all types of human solid cancer compared with normal tissue sample, whereas cytosolic ACC1 occurred equivalently in normal and cancer tissues (Fig 2A). These results suggest that metabolic reprogramming by transcriptional suppression of mitochondrial ACC2 plays an important role in the pathophysiology of human cancer. To assess potential association between Snail and ACC2, we further analyzed the transcript levels of those genes in breast cancer TCGA according to breast cancer subtypes and p53 tumor suppressor status. We found that (1) ACC2

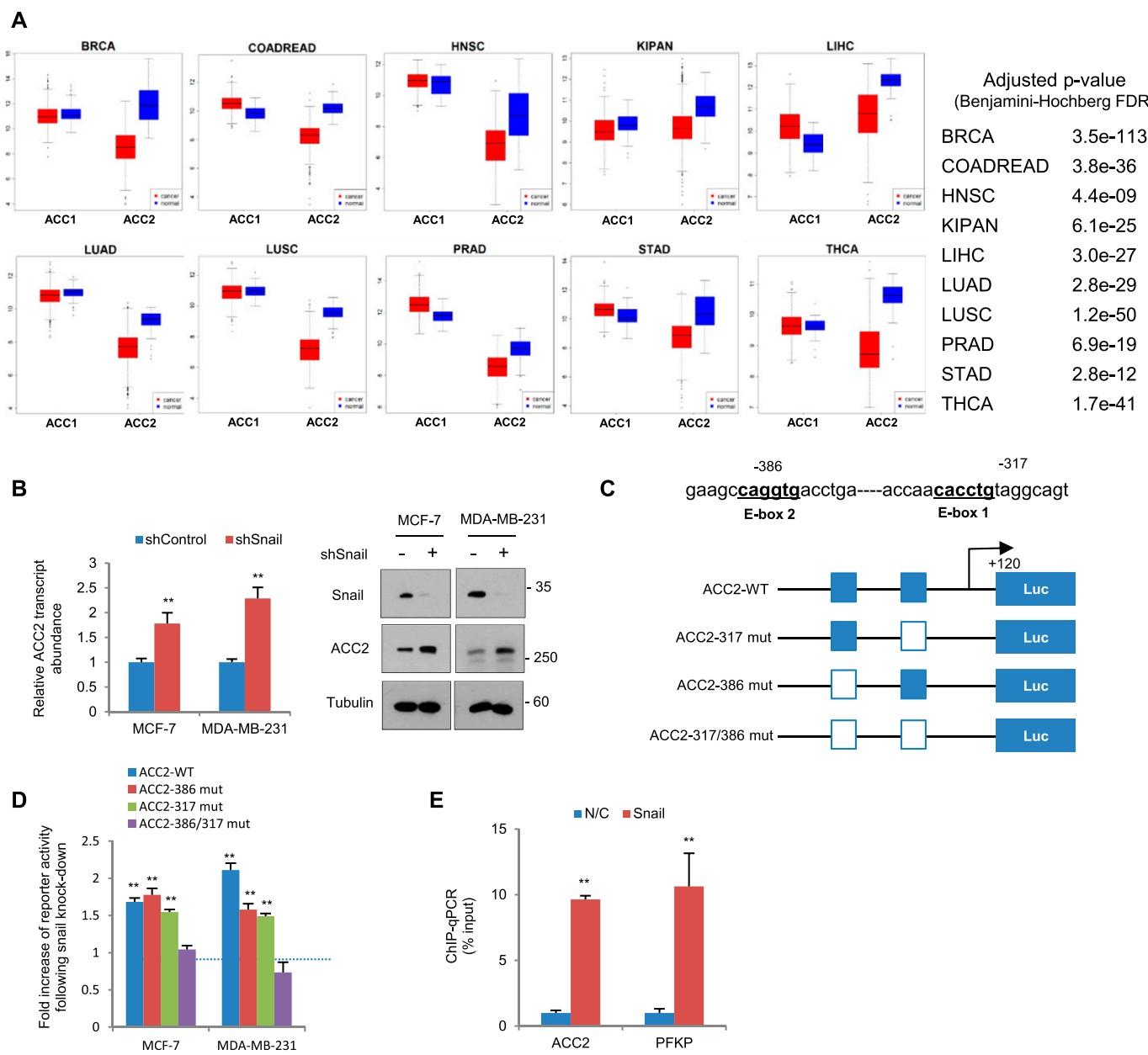

**Figure 2. Transcript abundance of mitochondrial ACC2 is suppressed in human cancer samples as a target of Snail repressor.**
**(A)** ACC2 (ACACB), but not ACC1 (ACACA), transcript abundance in tumor tissue (red) was suppressed compared with adjacent normal tissue (blue) in various cancer types. The Cancer Genome Atlas dataset included breast cancer (BRCA), colorectal adenocarcinoma (COADREAD), head and neck squamous cell carcinoma (HNSC), pan-kidney cohort (KIPAN), liver hepatocellular carcinoma (LIHC), lung adenocarcinoma (LUAD), lung squamous cell carcinoma (LUSC), prostate adenocarcinoma (PRAD), stomach adenocarcinoma (STAD), and thyroid carcinoma (THCA). The adjusted *P*-value of various cancer types were determined by the false discovery rates (FDRs) (Benjamini–Hochberg) method (right panel). **(B)** Relative transcript (left) and protein (right) abundance of ACC2 after knockdown of Snail (shSnail) in breast cancer cells (n = 3, means ± SD, *t* tests). **(C)** Schematic diagram showing positions of potential Snail-binding canonical E-boxes on the ACC2 proximal promoter region and its reporter constructs of wild type or mutant E-boxes. **(D)** Fold increase of reporter activities in combination with wild type or mutated ACC2 promoter following shRNA-mediated Snail knockdown compared with each control shRNA in breast cancer cells (n = 3, means ± SD, *t* tests). **(E)** ChIp-enriched DNA was determined by qRT-PCR using specific primers complementary to the promoter region containing E-box of ACC2 and a positive control PFKP (n = 3, means ± SD, *t* tests).

and Snail are inversely correlated in breast cancer tissue, (2) ACC2 and Snail transcript abundance were closely related to p53 status in clinical samples, and (3) ACC2 was down-regulated in triple-negative breast cancer and ERBB2-amplified subtypes (Fig S2), suggesting that Snail and ACC2 are specifically associated with breast cancer subtypes and p53 status.

Given our notion regarding the critical function of Snail repressor in FAO, we next examined whether ACC2 is a downstream target of Snail repressor. When we knocked down endogenous Snail and evaluated the ACC2 abundance, the transcripts and protein abundance of ACC2 were consistently increased by loss of Snail, whereas ACC1 transcript levels were minimally affected (Figs 2B and

S3A). Conversely, overexpression of Snail suppressed transcripts and protein abundance of ACC2 (Fig S3B), suggesting that ACC2 may be a downstream target of Snail repressor, similar to E-cadherin and phosphofructokinase, platelet (PFKP) (Cano et al, 2000; Kim et al, 2017). Indeed, the proximal ACC2 promoter region harbors two Snail-binding canonical E-boxes (CAGGTG), and we could clone the promoter region of ACC2 into the firefly luciferase reporter vector with mutations of the putative Snail-binding sites individually or in combination (Fig 2C). The ACC2 reporter activity increased when we silenced endogenous Snail in breast cancer cells, whereas specific mutation of predicted binding sites relieved the ability of Snail to suppress reporter activity (Fig 2D). Overexpression of Snail suppressed promoter activity in an E-box–dependent manner (Fig S3C). Upon chromatin immunoprecipitation (ChIP) assay to determine whether Snail locates a repressor complex in ACC2 promoter, we found that a DNA fragment containing E-boxes from ACC2 can be amplified from the immunoprecipitated genomic DNA samples from antibody of endogenous Snail (Fig 2E), indicating that ACC2 is a transcriptional target of EMT inducer Snail.

### Suppression of ACC2 promotes FAO, leading to cancer pro-survival

Malonyl-CoA catalyzed by ACC2 is a potent endogenous inhibitor of CPT1, resulting in suppression of FAO (Foster, 2012). Given the Snail-dependent malonyl-CoA level and ACC2 suppression, we next examined the roles of ACC2 in FAO in terms of cancer cell pro-survival. Consistent with the well-known role of ACC2, the malonyl-CoA levels were closely correlated with ACC2 abundance in breast cancer cells (Figs 3A and S4A). As the malonyl-CoA–mediated CPT1 regulatory axis has been firmly established (Foster, 2004, 2012), CPT1 activities followed by FAO activities are known to be inversely correlated with ACC2 abundance and malonyl-CoA levels (Figs 3B and C and S4B). Suppression of ACC2 with two independent set of shRNA largely augmented ATP levels in MCF-7 and MDA-MB-231 cells, especially under glucose starvation (Figs 3D and S4C), whereas overexpression of ACC2 had the opposite effect (Fig S4D). The anabolic FAS and catabolic FAO are mutually exclusive, FAS mainly consuming intracellular NADPH in cells (Fan et al, 2014). Indeed, the fall of malonyl-CoA and activation of FAO by knockdown of ACC2 increased NADPH level in breast cancer cells (Fig 3E), indicating that ACC2 abundance is critically important for NADPH homeostasis as well as for the production of cellular ATP. Collectively, loss of ACC2 abundance in human cancer largely increased pro-survival function and clonogenic capacity induced by glucose starvation (Fig 3F and G), whereas overexpression of ACC2 largely attenuated clonogenic potential in vitro (Fig S4E). To examine whether metabolic reprogramming regulated by ACC2 plays a role in tumor initiation and metastatic progression, we next performed an in vivo experiment. Indeed, induction of ACC2 in MDA-MB-231 cells largely suppressed the in vivo tumor initiation (Fig 3H). When these cells were injected into tail vein, increased ACC2 abundance suppressed the lung metastatic potential of MDA-MB-231-luc-D3H2LN cells phenocoping the loss of Snail (Fig 3I) (Ye et al, 2015; Ni et al, 2016; Kim et al, 2017). As anti-oxidative NADPH level is critically important for chemotherapeutic resistance (Kim et al, 2017), knockdown of

ACC2 conferred therapeutic resistance of breast cancer cells against paclitaxel (Fig S4F). To further determine the potential importance of ACC2 in a clinical setting, we analyzed the clinical outcomes in TCGA breast cancer samples. We noted an inverse relationship between ACC2 and overall survival in patients having wild-type, but not mutant p53 status (Fig 3J). These results support the importance of ACC2 transcript abundance in cancer initiation and progression, with therapeutic implications.

### Role of the Snail-ACC2 axis in metabolic reprogramming

Given the ability of Snail to potentiate cancer cell survival with increased ATP level, we sought to determine whether ACC2 constitutes a key element in the catabolic rewiring by Snail. To validate the Snail-ACC2 regulatory axis, we designed an experimental system in which regulation of ACC2 could rescue metabolic reprogramming of ATP production and FAO by Snail. Indeed, falling ATP levels due to loss of Snail, especially in glucose-starved condition, were rescued by knockdown of ACC2 (Fig 4A). Conversely, overexpression of ACC2-attenuated ATP levels increased by Snail (Fig S5A). Furthermore, ACC2 abundance was critical for FAO enzymatic activities regulated by Snail (Figs 4B and S5B). Investigating the pro-survival function of ACC2 under metabolic stress, we found suppression of ACC2 sufficiently rescued clonogenic potential due to loss of Snail (Fig 4C), whereas overexpression of ACC2 had the opposite effect (Fig S5C). Consistent with previous observations (Jeon et al, 2012; Kim et al, 2017), knockdown of Snail decreased NADPH levels and loss of ACC2 rescued the role of Snail repressor, whereas overexpression of ACC2 did the opposite (Figs 4D and S5D). These observations indicate that transcriptional suppression of ACC2 in human cancer plays a critical role during Snail-mediated catabolic metabolism of NADPH and ATP.

### Pharmacological inhibition of PPP and CPT1 attenuates cancer progression

We have recently shown that Snail increases PPP flux via suppression of PFKP, resulting in NADPH production to potentiate cancer cell survival under metabolic stress (Kim et al, 2017). In this study, we further report that Snail augments FAO by suppression of ACC2, not only providing essential ATP under glucose-starved condition but also suppressing NADPH consumption. Given the ability of Snail to increase FAO via suppression of ACC2 and subsequent activation of CPT1, we hypothesized a critical role of Snail in providing catabolic pro-survival function of cancer cells through NADPH homeostasis and ATP generation (Fig 5A). Because glucose-6-phosphate dehydrogenase (G6PD) is a gatekeeper of PPP, our observations suggest that G6PD and CPT1 can be used to therapeutically target the metabolic reliance of human cancer on PPP and FAO. To test this notion, we interrupted Snail-mediated metabolic reprogramming by overexpression of ACC2 and knocked down G6PD, then measured ATP levels. Indeed, the intervention in FAO together with PPP significantly depleted ATP levels in breast cancer cells (Fig 5B), supporting the notion that metabolic inhibition of PPP and FAO constitutes a novel therapeutic approach for human cancer.

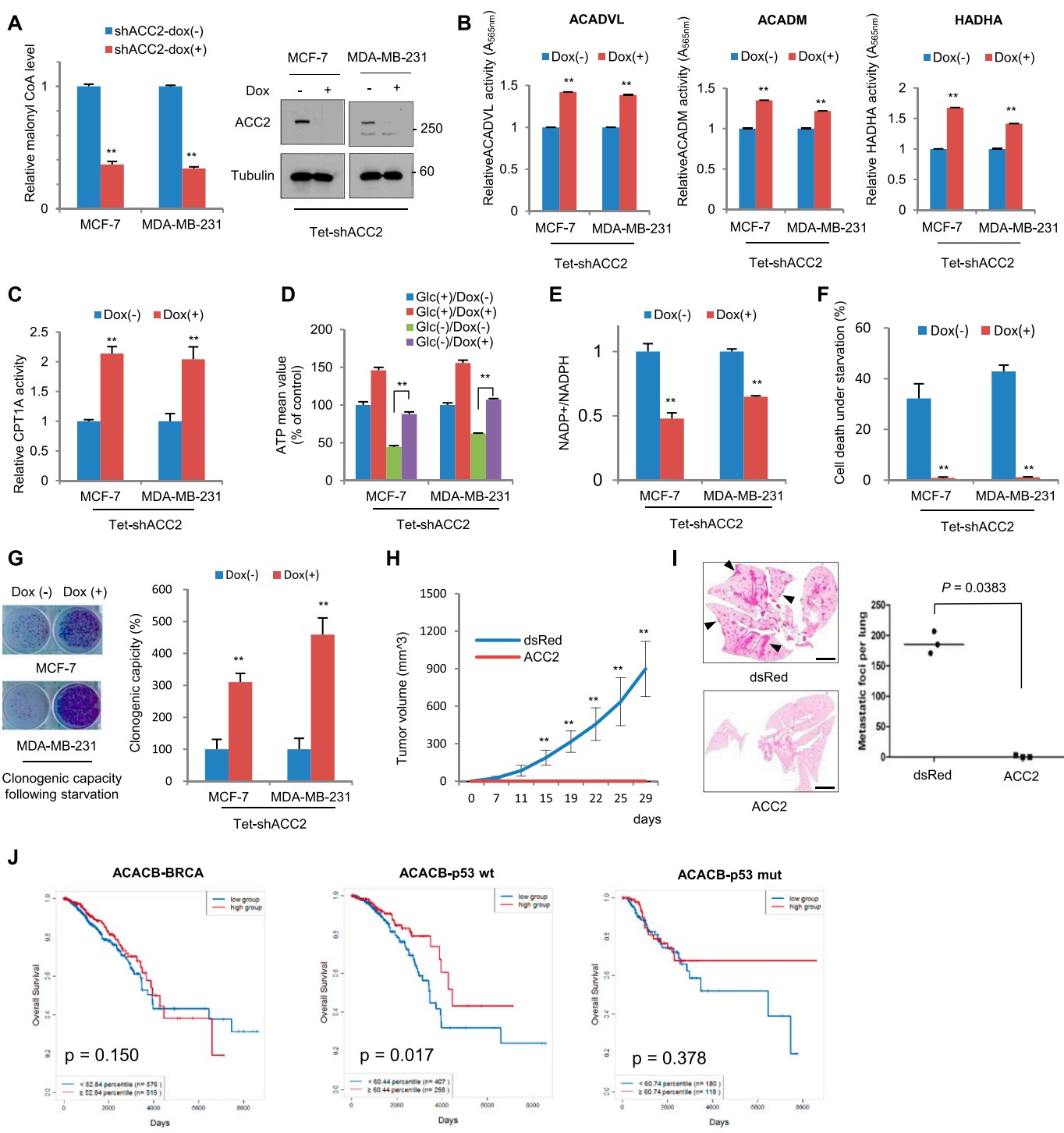

**Figure 3. Suppression of ACC2 increases CPT1 activity and fatty acid oxidation, providing pro-survival under metabolic stress.**
**(A)** Relative malonyl-CoA abundance according to ACC2 knockdown (left). ACC2 abundance was determined by immunoblot analysis after transduction of inducible shRNA (right). The double asterisks denote $P < 0.01$, one asterisk denoting $P < 0.05$ (n = 3, means ± SD, $t$ tests). **(B)** Fatty acid oxidation enzymatic activities under glucose-starved condition (0.5 mM) were determined after inducible knockdown (Dox+) of ACC2 compared with control (Dox−). **(C)** Knockdown of ACC2 increases CPT1 activity in breast cancer cells (n = 3, means ± SD, $t$ tests). **(D)** ACC2 was knocked down by treatment of doxycycline (Dox) for 48 h, and ATP levels were measured with glucose (Glc+) or glucose-starved (Glc−) condition (n = 3, means ± SD, $t$ tests). **(E)** ACC2 was knocked down by treatment of doxycycline (Dox) for 48 h, and NADP+/NADPH ratio was determined (n = 3, means ± SD, $t$ tests). **(F)** Cells were cultured under glucose-starved condition for a 48-h period, and cell death was quantitated by trypan blue exclusion assay (n = 3, means ± SD, $t$ tests). **(G)** Clonogenic survival assay of cancer cells after glucose starvation for a 72-h period. Colonies of more than 50 cells were counted after crystal violet staining (n = 5, means ± SD, $t$ tests). **(H)** MDA-MB-231 cells (1 × 10⁶) expressing dsRed (n = 3) or ACC2 (n = 3) were injected orthotopically into the mammary fat pads of nude mice. Tumor initiation and volume were monitored biweekly. Results are shown as means and SEM. Asterisks, $P < 0.01$ compared with the control by

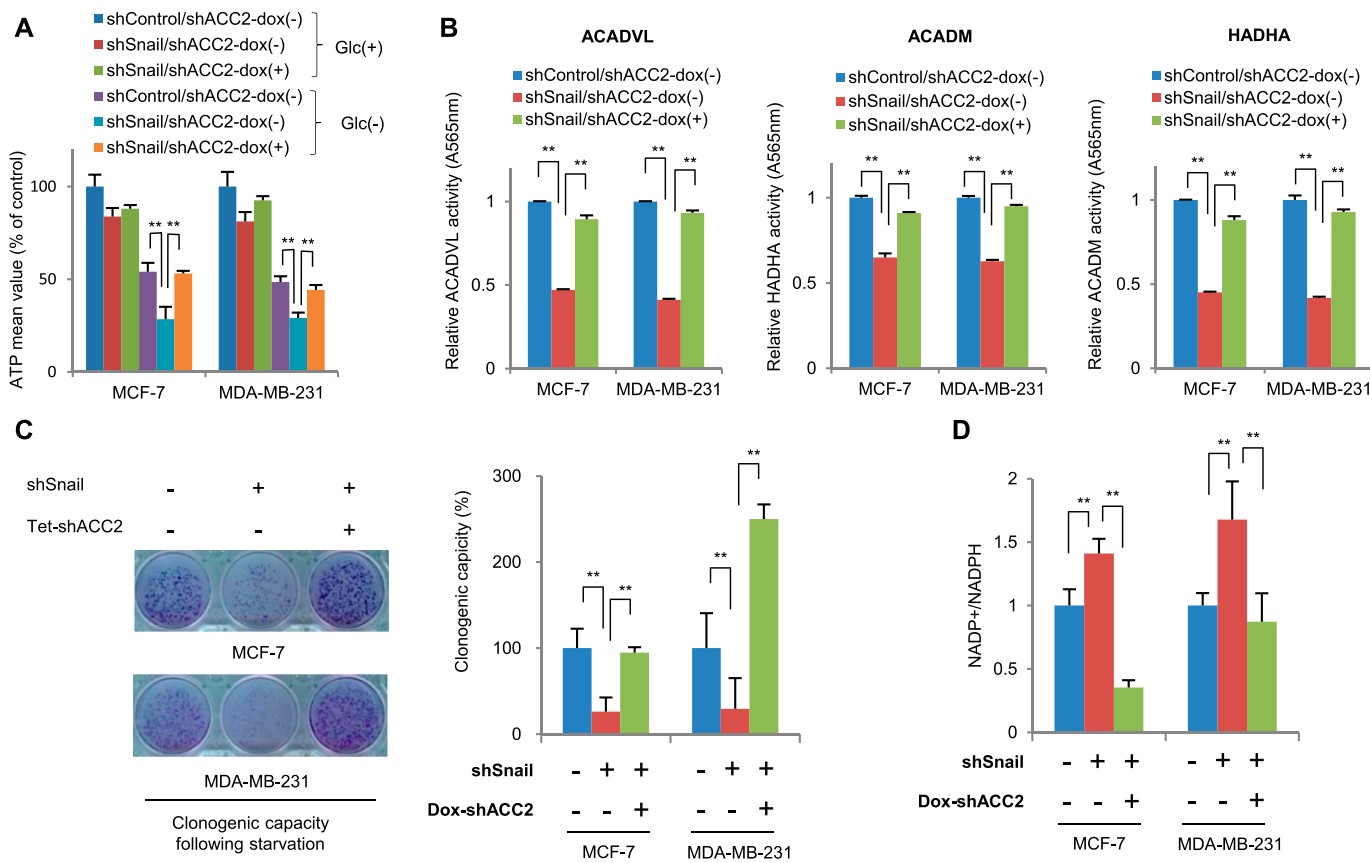

**Figure 4. Snail-ACC2 axis controls fatty acid oxidation and NADPH homeostasis.**
**(A)** Relative ATP level in breast cancer cells transduced with shRNA control (shControl) or with Snail shRNA (shSnail). ACC2 was knocked down by treatment with doxycycline (Dox+) for 48 h in Snail shRNA cells under either nourished condition (Glc+) or glucose-starved condition (Glc−). The double asterisks denote $P < 0.01$, one asterisk denoting $P < 0.05$ (n = 3, means ± SD, t tests). **(B, C, D)** Inducible knockdown of ACC2 rescued metabolic reprogramming by lack of Snail. The fatty acid oxidation activity (B), clonogenic capacity (C), and NADP+/NADPH ratio (D) after glucose starvation were measured. Data are means ± SD from n = 3 for (B, C), from n = 5 (D).

For a pharmaceutical approach against these metabolic targets, we chose dehydroepiandrosterone (DHEA) and etomoxir (ETX) as clinically available G6PD and CPT1 inhibitors, respectively. Interestingly, DHEA or ETX alone minimally affected NADPH and ATP levels in physiologic glucose concentration, whereas in combination, they significantly suppressed NADPH and ATP levels in breast cancer cells (Fig 6A). Examining real-time cell growth, we found the combination of DHEA and ETX significantly suppressed cell proliferation under physiologic glucose condition (Fig 6B). Examining clonogenic potential under metabolic stress, the dual pharmacological inhibition synergistically and dramatically suppressed the clonogenic potential of breast cancer cells (Fig 6C); although clinically available, DHEA is an uncompetitive G6PD inhibitor. To support our combinatorial approach, we next used CB83 as a novel selective inhibitor of G6PD (Preuss et al, 2013). To predict the binding mode for CB83 to the substrate binding site of hG6PD, a molecular docking simulation was performed. Docking pose analyses revealed that the benzene ring of the CB83 is embedded in a hydrophobic pocket, contributing to the complex formation; moreover, the sulfonamide and hydroxyl group present in the CB83 additionally formed hydrogen bonds with the lysine or aspartic acid side chain: Lys205, Asp258, and Lys360 (Fig S6A). These results indicate that the CB83 compound strongly interacts with the NADP+ substrate-binding site of hG6PD and show that the docked pose of this compound adopts conformations similar to those observed in the crystal structure of G6P-bound hG6PD (Kotaka et al, 2005). Indeed, CB83 combined with ETX had similar metabolic effects in terms of NADPH, ATP, cell growth, and clonogenic potential in breast cancer cells (Fig S6B–D). These results indicate that a pharmaceutical approach against PPP and FAO could efficiently intervene in metabolic reprogramming during Snail-mediated EMT.

Mann–Whitney test. **(I)** The MDA-MB-231-luc-D3H2LN cells (5 × 10$^5$ cells) either of control (dsRed, n = 3) or of ACC2 overexpression (ACC2, n = 3) were injected intravenously into tail veins of immunodeficient mice. The number of lung metastatic nodules at day 28 was counted under microscopic examination (left). Whole-field images of representative lungs showed the median value for each group. Statistical significance was determined by Mann–Whitney test (right). Arrows indicate metastatic tumor foci in mouse lung. Scale bar, 2 mm. **(J)** Kaplan–Meier survival graphs for all patients with breast cancer (left) or for those with wild-type p53 (middle) or p53 mutant (right) cancers, on the basis of ACC2 mRNA transcript abundance at an optimal threshold indicated by percentile numbers. Samples with decreased ACC2 expression are represented with blue lines. A log-rank test was used to calculate statistical significances.

**A**

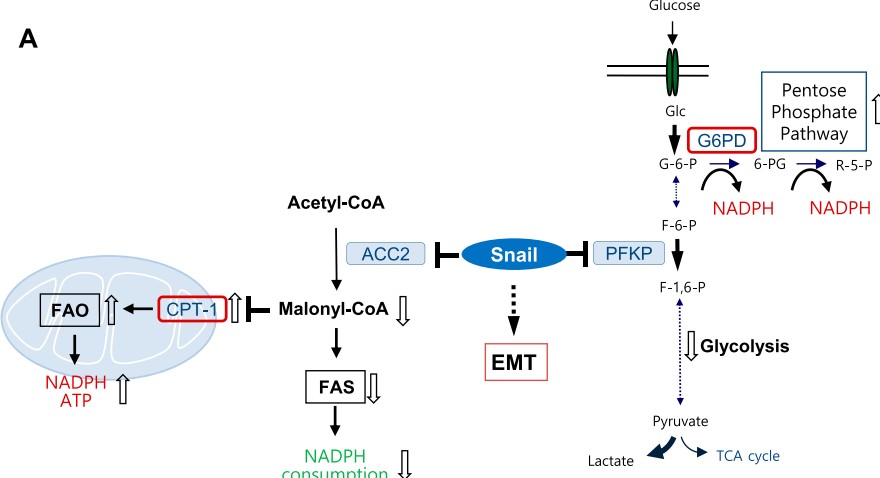

**Figure 5. Snail augments catabolic metabolism via activation of pentose phosphate pathway and fatty acid oxidation (FAO).**
**(A)** A schematic diagram depicting a potential catabolic mechanism by which the Snail regulates pentose phosphate pathway flux and FAO activity. Open arrows denote metabolic outcomes of catabolic metabolism regulated by increased Snail abundance in cancer cells. **(B)** Relative ATP levels (left) and immunoblot (right) in breast cancer cells according to ACC2 expression in combination with inducible glucose-6-phosphate dehydrogenase (G6PD) knockdown (Dox) under physiologic glucose concentration (5.5 mM, n = 3, means ± SD, t tests).

**B**

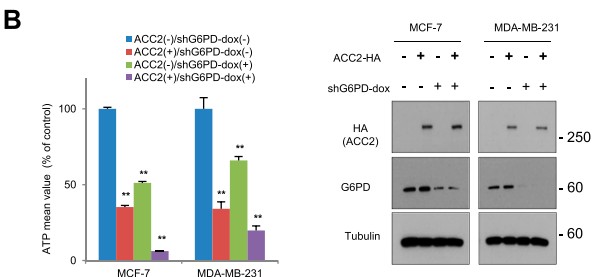

To test the in vivo therapeutic potential of this combined pharmaceutical approach, we next used a well-established xenograft model with MDA-MB-231 breast cancer cells. Although DHEA or ETX alone minimally suppressed in vivo tumor growth, their combination significantly suppressed tumorigenic potential as well as tumor growth (Fig 6D). We found no obvious systemic toxicity with combined pharmaceutical administration, including body weight (Fig S6E). Last, we further assessed the pharmacologic use of DHEA and ETX in a lung metastasis model. Indeed, the inhibition of PPP and CPT1 significantly suppressed the lung metastatic potential of MDA-MB-231-luc-D3H2LN cells (Fig 6E). These results indicate that pharmacologic inhibition of PPP and CPT1 may provide a therapeutic advantage for breast cancer patients.

## Discussion

Whereas glycolysis, known as the Warburg effect, has been known to constitute an important metabolic aspect of cancer cells for several decades, aerobic glycolysis is now widely accepted as the glucose metabolism for production of biomass, such as amino acids, nucleotides, and fatty acids, supporting the anabolic process of cancer cell proliferation (Vander Heiden et al, 2009; Cha et al, 2015). However, cancer cells repeatedly face metabolic stresses in the microenvironment such as limited glucose supply during the natural history of cancer progression (Aktipis et al, 2013). Interestingly, cancer cells in clinical tissues maintain their ATP level despite lack of glucose from

blood vessels (Walenta et al, 2003; Hirayama et al, 2009), suggesting a source of ATP other than aerobic glycolysis.

Anabolic FAS and catabolic FAO constitute mutually exclusive fatty acid metabolisms determined by the conversion of acetyl-CoA into malonyl-CoA by ACCs (Foster, 2004, 2012). Although malonyl-CoA in FAS is mainly involved in the elongation of fatty acids via fatty acid synthase (FASN), it also regulates CPT1 activity resulting in fatty acid transport into mitochondria (Foster, 2004, 2012). Therefore, malonyl-CoA levels determine the switch between FAS and FAO by inhibiting mitochondrial CPT1, a rate-limiting step in FAO (McGarry et al, 1978; Foster, 2004). Acetate has recently been highlighted as a source of acetyl-CoA in human cancer (Mashimo et al, 2014; Liu et al, 2018), although the regulatory role of acetyl-CoA on fatty acid metabolism is not well understood. Because the FAS largely consumes intracellular NADPH (Fan et al, 2014), reciprocal regulation of FAS and FAO by malonyl-CoA plays a critical role in ATP generation as well as NADPH homeostasis. Indeed, ACC activity and abundance are tightly regulated in cells, including in human cancer. For example, AMPK (AMP-activated protein kinase) phosphorylates the ACCs, thereby inhibiting synthesis of malonyl-CoA and subsequently activating CPT1 activity and FAO for ATP generation (Foster, 2004). Here, we found that Snail repressor regulates transcriptional levels of ACC2, thereby suppressing malonyl-CoA and activating CPT1 activity. Moreover, transcript abundance of mitochondrial ACC2, but not of cytosolic ACC1, was globally down-regulated in many types of human cancer samples compared with adjacent normal tissue, indicating that transcriptional repression as well as phosphorylation-mediated enzymatic inhibition of ACC2 comprises important metabolic rewiring in human solid cancer.

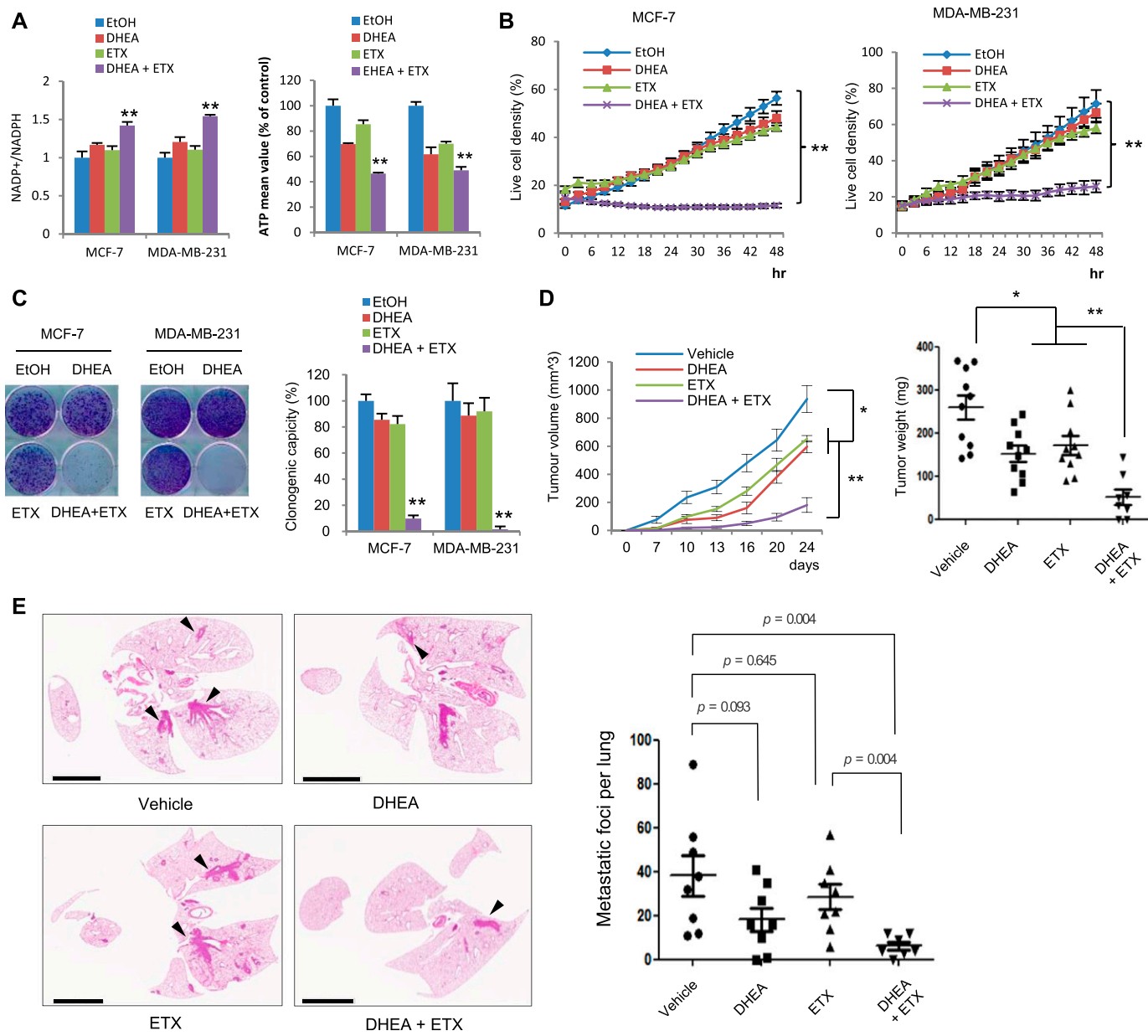

**Figure 6. Combined pharmacological inhibition of pentose phosphate pathway and CPT1 synergistically suppresses cancer progression.**
**(A)** A glucose-6-phosphate dehydrogenase inhibitor DHEA (20 μg/ml) in combination with CPT1 inhibitor etomoxir (ETX, 200 μM) was treated for a 16-h period. The NADP+/NADPH ratio (left) and ATP levels (right) were determined from breast cancer cells under physiologic glucose concentration (5.5 mM). The double asterisks denote $P <$ 0.01, one asterisk denoting $P < 0.05$ (n = 3, means ± SD, t tests). **(B)** Live cell density of breast cancer cells after treatment of DHEA (20 μg/ml) in combination with etomoxir (ETX, 200 μM) for a 48-h period under physiologic glucose concentration (n = 5, means ± SD, t tests). **(C)** Clonogenic survival of breast cancer cells under glucose-deprived condition (0.5 mM) in combination with DHEA (6.25 μg/ml) and/or etomoxir (ETX, 100 μM) followed by refreshment of normal culture medium (left). The colony number (right) was determined by stereomicroscopic examination as described in the Materials and Methods section (n = 5, means ± SD). **(D)** MDA-MB-231 cells (1 × 10^6) were orthotopically injected into the mammary fat pads of either vehicle (n = 10), or oral administration of DHEA (100 mg/kg, n = 10), or intraperitoneal administration of etomoxir (ETX, 50 mg/kg, n = 10), or a combination of DHEA and ETX (n = 9). The drugs were given five times a week and tumor growth was measured twice a week (means ± SEM). Two asterisks denote $P <$ 0.01 by Mann–Whitney test. **(E)** Lung metastasis by tail vein injection of MDA-MB-231-D3H2LN cells (5 × 10^5). The mice were administrated by either vehicle (n = 8), or oral administration of DHEA (100 mg/kg, n = 8), or intraperitoneal administration of etomoxir (ETX, 50 mg/kg, n = 8), or a combination of DHEA and ETX (n = 8). The number of lung metastatic nodules at day 28 was counted under microscopic examination (left), and statistical significance was determined by Mann–Whitney test. Whole-field images of representative lungs showing median value for each group (right). Arrows indicate metastatic tumor foci in mouse lung. Scale bar, 1 mm.

FAO is the most important catabolic metabolism by which fatty acid chains are serially shortened in the mitochondria to generate NADH, FADH2, and acetyl-CoA (Carracedo et al, 2013). In a highly efficient process, the complete oxidation of a C16-palmitate molecule yields 106 molecules of ATP. Although glycolysis is largely regarded as a metabolic source of ATP in cancer, the contribution of FAO as a source of ATP, especially in a glucose-limited microenvironment, is not yet clearly understood.

It should be noted that ATP generation is tightly connected with NADPH homeostasis because oxidative phosphorylation in the redox reactions inevitably produces reactive oxygen species. Thus, generation of ATP and reductive NADPH are essential for cancer cells to survive in a glucose-starved environment or to gain therapeutic resistance (Ramanathan et al, 2005; Schafer et al, 2009). While the Snail repressor has long been noted as a phenotypic EMT-inducer, recent observations indicate that Snail-mediated EMT comprises a wide range of cancer cell behavior, such as therapeutic resistance and stemness (Wolf, 2014). We reported that EMT-inducer Snail increases PPP flux by suppression of PFK-1, a gatekeeper of glycolytic flux converting fructose 6-phosphate (F6P) into fructose 1,6-bisphosphate (F1,6BP) (Kim et al, 2017; Cho et al, 2018). In this light, a recent report regarding Snail-mediated suppression of fructose-1,6-bisphosphatase (FBP1) in triple-negative breast cancer is of special interest because loss of FBP1 suppresses endergonic gluconeogenesis from F1,6BP to F6P that is the reverse enzymatic reaction of PFK-1 (Dong et al, 2013). It should be noted that PPP is tightly connected and coordinated with glycolysis and that inhibition of the glycolytic pathway by suppression of PFK-1 or pyruvate kinase activates PPP flux resulting in cancer progression (Anastasiou et al, 2011; Kim et al, 2017). Given the well-known role of bi-directional transketolase (TKT) and transaldorase (TALDO1) interconverting glyceraldehyde-3-phosphate in glycolysis and ribose-5-phosphate in PPP, loss of FBP1 by Snail also provides concomitant generation of NADPH and ATP as well as a building block in a variety of biosynthesis via nonoxidative branch of the PPP (Berg et al, 2012; Cho et al, 2018). Therefore, the interconversion of glycolytic glyceraldehyde-3-phosphate/F6P and ribose-5-phosphate explains the paradoxical suppressor function of Snail on glycolysis and gluconeogenesis, suggesting the existence of a different metabolic circuit according to cancer subset or oncogenic activation. In turn, EMT-inducer Snail increases PPP-dependent glucose metabolism rather than glycolytic interconversion between F6P and F1,6BP, a major rate-limiting step in glycolysis. In this study, we show that loss of ACC2 by Snail increases ATP and NADPH levels, resulting in pro-survival of cancer cells under metabolic stress. However, little is known about such catabolic metabolic reprogramming in certain metabolic circuits and fatty acid metabolism in special contexts. Further study is required to delineate the role of specific contexts, such as oncogenic activation and cancer subtypes, in Snail-mediated suppression of glycolysis or gluconeogenesis in metabolic advantages supporting breast cancer progression.

The Snail repressor is critically required for metastatic progression and plays a critical role in catabolic metabolism under glucose-limited condition (Ye et al, 2015). Therefore, targeting catabolic metabolism with clinically available drugs may provide therapeutic options for cancer patients, especially in the advanced stage. Recent studies have shown that CPT1 is overexpressed in various types of human cancer, making it a metabolic target (Camarda et al, 2016; Qu et al, 2016; Kim, 2018). Given the role of Snail repressor in metastatic cancer progression and metabolic rewiring to provide NADPH and ATP, we attempted combinatorial approaches to inhibit PPP and FAO. G6PD is an interesting metabolic target for cancer therapeutics, and DHEA is widely used for PPP inhibition (Fang et al, 2016; Cho et al, 2018). Indeed, knockdown of G6PD or DHEA increased oxidative stress in cancer cells, resulting in

decreased migratory potential and increased susceptibility to cell death under various stresses (Schafer et al, 2009; Fang et al, 2016; Kim et al, 2017). Earlier epidemiological studies show that DHEA level and its sulfated metabolite are inversely related to breast cancer risk, suggesting a possible effect of DHEA against breast cancer (Zumoff et al, 1981). Moreover, DHEA inhibits tumor development and progression in many types of animal models (Boccuzzi et al, 1992; Di Monaco et al, 1997), although the clinical effectiveness of DHEA alone is limited because of a relatively high IC50 level to control G6PD activity and subsequent high oral doses.

The ETX is an irreversible small molecule inhibitor of CPT1 on the outer face of the inner mitochondrial membrane (Kruszynska & Sherratt, 1987). Although the ETX has been developed for type 2 diabetes and heart failure through clinical trials (Fillmore & Lopaschuk, 2013), it is now receiving attention as a potential repositioned therapeutic because FAO has emerged as a metabolic target of cancer, regardless of its off-target effect (Divakaruni et al, 2018; Lee et al, 2019). Given the reciprocal balance between FAS and FAO mediated by malonyl-CoA (Foster, 2004, 2012), inhibition of FAO induces ATP depletion as well as impairs NADPH homeostasis (Jeon et al, 2012; Fan et al, 2014). In this study, we provide experimental evidence that combinatory metabolic inhibition against FAO and G6PD inhibition with clinically available drugs successfully suppresses metastatic cancer progression.

# Materials and Methods

## Cell culture and immunoblot analysis

MCF-7 and 293 cells obtained from ATCC were routinely cultured in DMEM medium containing 10% FBS. MDA-MB-231 cells (a gift from G Mills) were cultured in an RPMI1640 with 5% FBS. Mycoplasma infection was tested regularly with a PCR-based kit (MP0040; Sigma-Aldrich). Cell lines were authenticated as described recently (Kim et al, 2017). The transfection was performed by Lipofectamine 2000 according to the manufacturer's protocol (Invitrogen). For FAS assay, 3T3-L1 preadipocytes (kindly provided by JW Kim of the Yonsei Univeristy College of Medicine) were used. Post-confluent 3T3-L1 cells for 2 d (from "day 0") were cultured in DMEM with 10% FBS, 0.5 mM 3-isobutyl-1-methylxanthine (IBMX; Sigma-Aldrich), 2 μg/ml dexamethasone (Sigma-Aldrich), and 1 μg/ml insulin (Roche) for 2 d. After 2 d, the media were changed with fresh DMEM including 10% FBS and 1 μg/ml insulin. This was repeated every 2 d, adipocyte being valid for up to 8 d from the date of culture start. On day 8, the lipid droplet formation could be confirmed by Oil Red O staining. To measure the exact content of lipid accumulation, Oil Red O was dissolved in 100% isopropyl alcohol, and its absorbance was detected at the 490 nm wavelength with a spectrophotometer. For the Western blot analyses, protein extracts were prepared in Triton X-100 lysis buffer. Antibodies against Snail (3895S, mouse monoclonal L70G2, 1:2,000; Cell Signaling Technology), ACC2 (HPA006554, rabbit polyclonal, 1:1,000; Sigma-Aldrich), G6PD (sc373887, mouse monoclonal G6, 1:1,000; Santa Cruz), Flag (F-3156, mouse monoclonal, 1:5,000; Sigma-Aldrich), and tubulin (LF-PA0146, rabbit polyclonal, 1:5,000; AbFrontier) were obtained from commercial vendors. Dehydroepiandrosterone (DHEA,

CAS 53-43-0) and etomoxir (ETX, CAS 828934-41-4) were purchased from Calbiochem and Sigma-Aldrich, respectively. CB83 (N-(4-hydroxynaphthalen-1-yl)-2,5-dimethylbenzenesulfonamide, C18H17NO3S, MW 327.4) was synthesized by the 4Chem Laboratory.

## Quantitative real-time PCR (qRT-PCR)

Total RNA was isolated using TRIzol reagent (Invitrogen) following the manufacturer's protocol. The SuperScript III synthesis kit (Invitrogen) was used to generate cDNA. qRT-PCR analysis for ACC1 and ACC2 transcripts was performed with an ABI-7300 instrument under standard conditions and SBGR mix (n = 3). The ΔCt value expression from each sample was calculated by normalizing with GAPDH. Primer specificity and PCR process were verified by dissociation curve after PCR reaction. The primer sequences for qRT-PCR were 5'-ctcttgaccctggctgtgtactag for ACC1 forward, 5'-tgagtg ccgtgctctggat for ACC1 reverse, 5'-tccgcggctataatgaaaacag for ACC2 forward, 5'-tcgtagtgggcttgctgaaa for ACC2 reverse, 5'-atgggtgtg aacccatgagaag for GAPDH forward, and 5'-agttgtcatggatgaccttgg for GAPDH reverse.

## Plasmids and RNA-mediated interference

Tetracycline-inducible Snail expression vector was generated with the pTRIPZ lentiviral system (Open Biosystems) by replacing red fluorescent protein with Flag-tagged Snail and lentivirus-mediated (pLL3.7-dsRed). Snail knockdown was reported recently (Kim et al, 2017). The Tet-pLKO-puro vector (#21915, obtained from Addgene) was used for an independent set of inducible shRNA knockdown of Snail or G6PD (Vega et al, 2004). Lentiviral ACC2 expression and shRNA vector were kindly provided by SM Jeon (Ajoo University) (Jeon et al, 2012). The target sequences of shRNA were 5'-gtggagctgattgtggacatt for shACC2; an independent set of siACC2 was purchased from Santa Cruz Biotechnology.

## Liquid chromatography–mass spectrometry measurement

For preparation of palmitate-BSA conjugate, sodium palmitate (1 mM, P9767; Sigma-Aldrich) in 150 mM NaCl was prewarmed and stirred at 70°C. Then, prewarmed fatty acid–free BSA solution (0.17 mM, 6:1 molar ratio palmitate:BSA) was added to palmitate solution and stirred for 1 h at 37°C, and the final volume adjusted to 1 mM palmitate:0.17 mM BSA. The MDA-MB-231 cells expressing inducible Snail-shRNA were treated with 13C-palmitate (100 $\mu$M) for 4 h, and the cell pellets were extracted by three times of freeze–thaw cycle using solvent system DW:MeOH:CHCl$_3$ (1:2:2). The mixtures were subsequently centrifuged at 14,000$g$ at 4°C for 10 min. The separated upper and lower layers were further evaporated by vacuum evaporator, and the samples were kept at –80°C until further analysis. The dried hydrophilic phase above was reconstituted in a mixture of DW:ACN (20 $\mu$l, 1:1 vol/vol) containing [U-13C]-glutamine as internal standard and then introduced into an liquid chromatography–mass spectrometry system. Metabolites were separated by Zic-pHilic column (150 × 2.1 mm, particle size: 5 $\mu$m; Merck) at 40°C using Acquity UPLC Waters coupled with Q Exactive Focus Hybrid Quadrupole-Orbitrap Mass Spectrometer (Thermo Fisher Scientific). The mobile phases were 10 mM ammonium carbonate (A) and

ACN (B) with the gradient as follows: 20% A from 0 to 2 min, then increasing gradually to 80% for 17 min, and kept there for 4 min. Set 20% A at 23.1 min was followed by 2 min equilibrium with a 0.15 ml/min flow rate. The Q Exactive Focus MS system equipped with a heated electrospray ionization (HESI-II) probe with the following settings: sheath gas = 40 ml/min; auxiliary gas = 10 ml/min, heated to 250°C; sweep gas = 2 ml/min; spray voltage = 2.5 kV, followed by capillary temperature at 256°C; and S-lens RF level = 100. The resolution was set at 70,000, and the AGC target was set at 1 × 10$^6$. Data were acquired in negative mode, and isotopologue peaks were extracted using Xcalibur ver. 2.8 (Thermo Fisher Scientific). The raw data of the results can be found in Supplemental Data 1.

## Metabolic analysis

The intracellular quantitation of NADPH and total NADP (NADPH and NADP+) were measured using the colorimetric method (K347; BioVision). The concentration of NADP+ was calculated by subtracting [NADPH] from [total NADP]. Quantitative enzymatic activities of ACADVL, ACADM, and HADHA in the mitochondrial FAO pathway were measured with a commercial colorimetric assay kit according to the manufacturer's protocol (ab118182; Abcam). CPT1 activity was measured with a commercial kit based on ELISA (OKEH00404; Aviva Systems Biology). Malonyl-CoA content was measured with a human malonyl-co enzyme A ELISA kit (MBS705079; MyBioSource) according to the manufacturer's specifications. ATP level was measured with an ATP colorimetric assay kit (K354; BioVision) in the absence or presence of glucose and FBS before starting preparation of seeded cells (1 × 10$^6$) according to the manufacturer's instructions. The oxygen consumption rates (OCRs) were measured using the Seahorse XF-24 instrument (Seahorse Bioscience) under standard conditions after the addition of 1 $\mu$M oligomycin, 0.25 $\mu$M FCCP, and 0.5 $\mu$M rotenone/antimycin A. Real-time measurements of the OCR in pmol per minute per $\mu$g per protein in adjusted base medium (L-glutamine 2.05 mM, D-glucose 2gL-1) were plotted over time before and after the addition of rotenone/antimycin A to specifically measure non-mitochondrial respiration. The difference in OCR between initial non-drug treatment and after addition of rotenone/antimycin A reflects the basal oxygen consumption by mitochondria. The OCR measurements were normalized with protein abundance of plated cells through BCA assay at a wavelength of 562 nm.

## ACC2 promoter cloning and reporter assay

The promoter region (–594 ~ –1 from the starting codon) of ACC2 was amplified from genomic DNA of MCF-7 cells and subcloned into pGL3 basic vector (Promega). E-box sequence 5'-CAGGTG was mutated to 5'-AAGGTA. To examine ACC2 promoter activity, the breast cancer cells were transfected with 100 ng of reporter vectors and 5 ng of pSV-Renilla expression vector. Luciferase and Renilla activities were measured using the dual-luciferase reporter system kit (Promega), the luciferase activity being normalized with Renilla activity. The results are expressed as the averages of the ratios of the reporter activities from triplicate experiments. For ChIP analysis, cells were cross-linked with 1% (vol/vol) formaldehyde at room temperature and suspended in a ChIP lysis buffer containing 10 mM

Hepes (pH 7.9), 0.5% NP-40, 1.5 mM MgCl₂, 10 mM KCl, 0.5 mM DTT, and protease inhibitor cocktail on ice (Kim et al, 2017). After centrifugation, the pellets were resuspended in buffer containing 20 mM Hepes (pH 7.9), 25% glycerol, 0.5% NP-40, 0.42 M NaCl, 1.5 mM MgCl₂, 0.2 mM EDTA, and protease inhibitors followed by extraction of nuclear proteins. Nuclear pellets were resuspended in IP buffer (1% Triton X-100, 2 mM EDTA, 20 mM Tris–HCl [pH 8.0], 150 mM NaCl, and protease inhibitors) and sonicated to break chromatin into fragments with an average length of 0.5–1 kb. The supernatants were incubated overnight with anti-Snail antibody or corresponding normal IgG. After incubation with protein G-agarose beads (Invitrogen) for 2 h at 4°C, the beads were extensively washed with lysis buffer. Bound DNA–protein complexes were eluted after an overnight incubation at 65°C in TE buffer. To amplify the ACC2 and PFKP promoter fragments as a positive control, the following primers were used for RT-PCR analysis: ACC2, forward 5'-attgcacatgtgacctcctg, reverse 5'-cctgtagatctggggtgtca; PFKP, forward 5'-ctagagcccccaacca-gagt, reverse 5'-gtgtgggcaggagcatctac. The results were normalized relative to input activities and presented as mean ± SD from three independent experiments.

### Cell death and clonogenic survival capacity

For the glucose starvation experiments, $5 \times 10^3$ cells were plated into six-well plates with normal culture medium a day before starvation. The cells were washed with PBS and cultured in glucose-free DMEM containing 10% FBS adjusted to ~0.5 mM glucose. The MDA-MB-231 cells and MCF-7 cells were starved for 48 and 72 h, respectively. Cell death induced by glucose starvation was measured by trypan blue exclusion assay. Separately, clonogenic survival was performed by exposing cells to glucose-starved condition for 72 h followed by further observation in normal culture medium for 14 d (Franken et al, 2006). After crystal violet (0.5% wt/vol) staining, colonies of more than 50 cells were counted under a stereomicroscope. The results of clonogenic assay are expressed as the ratios of the number of survival colonies compared with control. For clonogenic survival against paclitaxel treatment, $5 \times 10^3$ cells were plated into six-well plates with normal culture medium 48 h before paclitaxel treatment in the absence or presence of doxycycline for inducible expression of Snail or ACC2 shRNA. The cells were then cultured in paclitaxel-containing culture medium for 48 h followed by refreshment of normal culture medium for an additional 10–14 d to determine clonogenic survival. The number of colonies in five randomly chosen fields was determined under a high power stereomicroscope. Real-time cell growth was analyzed by JuLI Stage systems (NaoEn Teck) according to the manufacturer's instructions. Cell number was counted with an EVE automatic counter (NanoEnTek).

### Gene expression analysis of clinical samples

Publicly available mRNASeq data of human cancer samples including normal tissue (>30 samples) were downloaded (https://gdac.broadinstitute.org). The level 3 dataset (data version 2026_01_28) included breast cancer (BRCA, 1,100 cancer samples with 112 normal tissue samples), colorectal adenocarcinoma (COADREAD, 626 cancer with 51 normal), head and neck squamous cell carcinoma (HNSC, 522

cancer with 44 normal), pan-kidney cohort (KIPAN, 891 cancer with 129 normal), liver hepatocellular carcinoma (LIHC, 373 cancer with 50 normal), lung adenocarcinoma (LUAD, 517 cancer with 59 normal), lung squamous cell carcinoma (LUSC, 501 cancer with 51 normal), prostate adenocarcinoma (PRAD, 498 cancer with 52 normal), stomach adenocarcinoma (STAD, 415 cancer with 35 normal), and thyroid carcinoma (THCA, 509 cancer with 59 normal). The data links for the TCGA raw data are available in Supplemental Data 2. The illuminahiseq_rnaseqv2-RSEM_genes_normalized (MD5) was log₂ transformed, and the relative transcript abundance of ACC1 (ACACA) and ACC2 (ACACB) were compared using adjusted P-value (Benjamini–Hochberg). For an unsupervised hierarchical cluster analysis of ACC2 and Snail transcripts, Ward linkage method was used together with the Pearson distance for both sample and gene clustering. The statistical significance of ACC2 and Snail transcript according to the p53 mutational status and cancer subtype was determined by Tukey's HSD (honestly significant difference) test. To generate Kaplan–Meier plots according to ACC2 transcript abundance, clinical samples were grouped by p53 mutational status, and the plots were generated using the R package survival.

### Molecular modeling of CB83 compound and G6PD

Molecular docking calculations were performed using the AutoDock 4.2 programs to evaluate the binding mode of CB83 within a G6PD substrate binding site (PDB code 2BHL, https://www.rcsb.org/structure/2BHL) using an empirical free energy force field and rapid Lamarckian genetic algorithm search method. The maximum number of energy evaluations was set to 500,000, default values being used for the other parameters. The Lamarckian genetic algorithm was chosen to determine the best conformers in 50 independent trials of ligand. The lowest energy conformation in the most populated of the first five clusters was used for docking pose analysis. The structural figures were visualized using the PyMOL program.

### Xenograft and in vivo metastasis

All animal experiments were performed in accordance with the Institutional Animal Care and Use Committee of the Yonsei University and approved by the Animal Care Committee of the Yonsei University College of Dentistry and National Cancer Center Research Institute. Female athymic nude mice (6 wk old) were used for orthotopic xenograft assays into the mammary fat pads and lung metastasis assay. For loss of function study with ACC2 over-expression, MDA-MB-231 cells of control or experimental cells were harvested with trypsin treatment and injected orthotopically into the mammary fat pads ($1 \times 10^6$ per 0.1 ml of PBS). The tumor initiating capacity was measured twice a week using a digital caliper and the tumor volume was calculated with equation V (in mm³) = (a × b²)/2, where a is the longest and b is the shortest diameter. For lung metastasis assays, MDA-MB-231-luc-D3H2LN cells expressing control vector or ACC2 were injected into the lateral tail vein, detailed experimental conditions being described in each figure legend. Lung colonization was monitored and quantified using pathological examination at day 28. Mice were euthanized and perfused with 4% paraformaldehyde, and lungs were extracted for

paraffin-embedding. Paraffin-embedded sections were stained with routine hematoxylin and eosin and the number of lung metastatic nodules was counted under microscopic examination. For pharmacologic inhibition of tumorigenesis and lung metastasis, mice received vehicle or 50 mg/kg etomoxir dissolved in PBS (i.p.) or 100 mg/kg of DHEA (p.o.) or combination of etomoxir and DHEA after injection of MDA-MB-231 cells. The vehicle or drugs were administered five times a week for 3 wk for tumorigenesis and 4 wk for lung metastasis assay.

### Statistical analysis and reproducibility

All statistical analysis of reporter assay, RT-PCR, and soft agar assay was performed with two-tailed $t$ tests; data are expressed as means and SD. The double asterisks denote $P < 0.01$, one asterisk denoting $P < 0.05$. Statistical significance of animal experiments was determined using the Mann–Whitney test. No statistical method was used to predetermine sample size.

## Data Availability

The data that support the findings of this study are available from the corresponding author upon reasonable request.

## Supplementary Information

## ACKNOWLEDGEMENTS

We thank E Tunkle for preparation of the manuscript and KY Kim for statistical analysis. We also thank HG Kim at Avison Biomedical Research Center for technical assistance with the lung metastasis assay. This work was supported by grants from the National Research Foundation of Korea (NRF-2016R1E1A1A01942724, NRF-2017R1A2B3002241, NRF-2018M3A9E2022820, NRF-2018R1D1A1B07050744, and NRF-2019R1A2C2084535) funded by the Korea government (MSIP), a grant from the Korean Health Technology R&D Project, Ministry for Health and Welfare, Republic of Korea (HI17C2586), and a grant from the Yonsei University College of Dentistry Fund (6-2018-0006).

### Author Contributions

JH Yang: data curation and methodology.
NH Kim: conceptualization and data curation.
JS Yun: data curation.
ES Cho: data curation and funding acquisition.
YH Cha: conceptualization.
SB Cho: data curation and software.
S-H Lee: data curation.
SY Cha: data curation.
S-Y Kim: conceptualization.
J Choi: data curation.
T-TM Nguyen: data curation.
S Park: data curation.
HS Kim: conceptualization, funding acquisition, and project administration.
JI Yook: conceptualization and project administration.

### Conflict of Interest Statement

The authors declare that they have no conflict of interest.

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
