## [Reviewer comments · Life Science Alliance]

Life Science Alliance

Snail augments fatty acid oxidation by suppression of mitochondrial ACC2 during cancer progression

Jong Yook, Ji Hye Yang, Nam Kim, Jun Seop Yun, Eunae Cho, Yong Cha, Sue Bean Cho, Seon-Hyeong Lee, So Cha, Soo-Youl Kim, Jiwon Choi, Tin-Tin Manh Nguyen, Sunghyok Park, and Hyun Kim

DOI: <https://doi.org/10.26508/lisa.202000683>

Corresponding author(s): Jong Yook, Yonsei University College of Dentistry and Hyun Kim, College of Dentistry Yonsei University

Review Timeline:

Submission Date:	2020-02-19
Editorial Decision:	2020-04-28
Revision Received:	2020-05-07
Editorial Decision:	2020-05-18
Revision Received:	2020-05-20
Accepted:	2020-05-22

Transaction Report:

April 28, 2020

Re: Life Science Alliance manuscript #LSA-2020-00683-T

Prof. Jong In Yook
Yonsei University College of Dentistry
Department of Oral Pathology
50-1 Yonsei-ro, Seodaemun-gu
Seoul 120-752
Korea, Republic of

Dear Dr. Yook,

Thank you for submitting your manuscript entitled "Snail augments fatty acid oxidation by suppression of mitochondrial ACC2 during cancer progression" to Life Science Alliance. Please excuse the delay in getting back to you. It was very difficult to secure reviewers on your work and we also had to give them more time. In the end, only one reviewer submitted a report, which is mostly positive, while another reviewer refused to provide a full report given the unbalanced way you put your work into the context of the existing literature. You will find the reviewer report below and I also copy the constructive comments of the other reviewer at the end of this message (advisor comments).

Based on the input received, we decided that we can invite you to submit a revised version of your work to us. We would expect that you fully address the reviewer's comments. Furthermore, we'd expect that you include a much more complete and balanced discussion of the role of Snail in metabolism - including citing and discussing key papers that show that Snail increases glycolysis, and inclusion of either experimental data addressing the controversial findings or clearly discussing the discrepancies and the need to resolve them.

The typical timeframe for revisions is three months. We are aware that many laboratories cannot function fully during the current COVID-19/SARS-CoV-2 pandemic and therefore encourage you to take the time necessary to revise the manuscript to the extent requested above. We will extend our 'scooping protection policy' to the full revision period required. Please note that papers are generally considered through only one revision cycle, so strong support from the referee on the revised version is needed for acceptance.

Thank you for this interesting contribution to Life Science Alliance. We are looking forward to receiving your revised manuscript.

Sincerely,

B. MANUSCRIPT ORGANIZATION AND FORMATTING:

*****IMPORTANT:** It is Life Science Alliance policy that if requested, original data images must be made available. Failure to provide original images upon request will result in unavoidable delays in publication. Please ensure that you have access to all original microscopy and blot data images

before submitting your revision.***

Reviewer #2 (Comments to the Authors (Required)):

Summary

This paper provides mechanistic evidence that the transcription factor Snail, previously implicated in the epithelial to mesenchymal transition, enhances fatty acid oxidation (FAO) to promote cancer cell survival in glucose-starved conditions. Further studies suggest Snail expression leads to enhanced fatty acid transporter activity. This builds on the team's recent paper, Kim et al., 2017 Nat Comm, which focuses on Snail's ability to re-programme glucose metabolism by suppressing PKFP to facilitate cancer cell survival.

Main points and supporting data

The paper has a logical and thoughtful flow. Its main points seem well supported by a combination of pharmacological and genetic knockdown studies in vitro, analysis with xenograft models in vivo and some limited cancer patient data. The study focusses on breast cancer, except for some patient data that do not directly link to Snail; this should be made clearer in the Abstract, Discussion, etc. Overall, mechanistic insights are supported by rescue experiments, eg use of the CPT1 inhibitor ETX blocks palmitate-induced ATP recovery and rescue of Snail knockdown-mediated reprogramming by inducible ACC2 knockdown. I do not have expertise in fatty acid oxidation metabolism, but the general conclusions appear novel and interesting. There is a potential mechanism proposed for Snail's effect on the ACC2 promoter, but the authors do not flag this in the Abstract, which might have been useful.

It would have been preferable to see the western data performed in triplicate, so that statistical differences could be shown, but this may be difficult in the current difficult circumstances.

Additional issues that should be addressed

For the most part, the paper is clearly written. I have, however, identified a few places where the text needs changing and there are others, so the text should be carefully proof-read and edited prior to resubmission.

When first mentioned (page 7) the authors should explain what palmitate does.

Page 3. Highlight, and elsewhere: 'combinatorial' instead of 'combinational'.

Page 4, line 2 from top: replace 'had' with 'has'

Page 5, line 6 from bottom: replace 'cancer stemness' with 'cancer cell stemness'

Page 6, bottom line: specify what pharmacological strategy is for

Page 7, line 10 from top; Page 9, line 11 from top: rephrase 'Snail repressor'

Page 7, line 6 from bottom: replace 'treated' with 'administered'

Page 11, line 8 from top: I think 'leading to cancer pro-survival' could be phrased better

Page 16, line 10 from top: replace 'level' with 'levels'

Page 16, line 14 from top: rephrase 'constitute fatty acid metabolism'

Figure 1A legend (Page 33) mentions an immunoblot analysis, which is missing from the figure

Figure 3D legend (Page 36) 'knockdowned' should be replaced by 'knocked down'

Advisor comments:

- It is difficult to reconcile the data with previous knowledge on Snail and metabolism. The authors have already published that Snail inhibits glycolysis by repressing phosphofructokinase (PFK1), but several others have reported the opposite. Lu et al 2015 show that Snail increases glycolysis attenuating gluconeogenesis and Dong et al 2013 show that Snail represses fructose 1,6

biphosphatase (FBP1), therefore increasing glycolysis as well. - In this paper, the authors start by mentioning their previous data on glycolysis inhibition mediated by Snail and then in the discussion, they use Dong et al (inhibition of FBP1) to say that it is of special interest. They see that Snail enhances Fatty acid biosynthesis (FAO) they now propose that maybe this compensates for FBP1 Loss. Snail inhibiting PFK and FBP is very difficult to understand and they use one paper or another. - The authors use references in many instances all throughout the paper that do not fit with what is stated in the text.

Response to Reviewers (LSA-2020-00683-T)

We appreciate the reviewers' constructive criticism and helpful comments on our manuscript. Our point-by-point responses to the reviewers are provided in bold below.

=====

Reviewer #2 (Comments to the Authors) :

Summary

This paper provides mechanistic evidence that the transcription factor Snail, previously implicated in the epithelial to mesenchymal transition, enhances fatty acid oxidation (FAO) to promote cancer cell survival in glucose-starved conditions. Further studies suggest Snail expression leads to enhanced fatty acid transporter activity. This builds on the team's recent paper, Kim et al., 2017 Nat Comm, which focuses on Snail's ability to re-programme glucose metabolism by suppressing PKFP to facilitate cancer cell survival.

Main points and supporting data

The paper has a logical and thoughtful flow. Its main points seem well supported by a combination of pharmacological and genetic knockdown studies in vitro, analysis with xenograft models in vivo and some limited cancer patient data. The study focusses on breast cancer, except for some patient data that do not directly link to Snail; this should be made clearer in the Abstract, Discussion, etc. Overall, mechanistic insights are supported by rescue experiments, eg use of the CPT1 inhibitor ETX blocks palmitate-induced ATP recovery and rescue of Snail knockdown-mediated reprogramming by inducible ACC2 knockdown. I do not have expertise in fatty acid oxidation metabolism, but the general conclusions appear novel and interesting. There is a potential mechanism proposed for Snail's effect on the ACC2 promoter, but the authors do not flag this in the Abstract, which might have been useful.

It would have been preferable to see the western data preformed in triplicate, so that statistical differences could be shown, but this may be difficult in the current difficult circumstances.

Authors' Response) We appreciate the reviewer's helpful comment on our manuscript. We have added the comments on breast cancer in the Abstract and Discussion, and on transcriptional repression in Abstract.

Additional issues that should be addressed

For the most part, the paper is clearly written. I have, however, identified a few places where the text needs changing and there are others, so the text should be carefully proof-read and edited prior to resubmission.

When first mentioned (page 7) the authors should explain what palmitate does.

Page 3. Highlight, and elsewhere: 'combinatorial' instead of 'combinational'.

Page 4, line 2 from top: replace 'had' with 'has'

Page 5, line 6 from bottom: replace 'cancer stemness' with 'cancer cell stemness'

Page 6, bottom line: specify what pharmacological strategy is for

Page 7, line 10 from top; Page 9, line 11 from top: rephrase 'Snail repressor'

Page 7, line 6 from bottom: replace 'treated' with 'administered'

Page 11, line 8 from top: I think 'leading to cancer pro-survival' could be phrased better

Page 16, line 10 from top: replace 'level' with 'levels'

Page 16, line 14 from top: rephrase 'constitute fatty acid metabolism'

Figure 1A legend (Page 33) mentions an immunoblot analysis, which is missing from the figure

Figure 3D legend (Page 36) 'knockdowned' should be replaced by 'knocked down'

Authors' Response) We appreciate the reviewer's careful comments on our manuscript. We have corrected the mistakes in the revised manuscript following the reviewer's suggestions.

Advisor comments:

It is difficult to reconcile the data with previous knowledge on Snail and metabolism. The authors have already published that Snail inhibits glycolysis by repressing phosphofruktokinase (PFK1), but several others have reported the opposite. Lu et al 2015 show that Snail increases glycolysis attenuating gluconeogenesis and Dong et al 2013 show that Snail represses fructose 1,6 biphosphatase (FBP1), therefore increasing glycolysis as well. In this paper, the authors start by mentioning their previous data on glycolysis inhibition mediated by Snail and then in the discussion, they use Dong et al (inhibition of FBP1) to say that it is of special interest. They see that Snail enhances Fatty acid biosynthesis (FAO) they now propose that maybe this compensates for FBP1 Loss. Snail inhibiting PFK and FBP is very difficult to understand and they use one paper or another.

Authors' Response) We understand the reviewer's concern regarding Snail's role in glycolysis and gluconeogenesis. In this study, we focused on Snail's role in metabolic reprogramming, especially for the efficient supply of essential catabolic metabolites (ATP and NADPH) in a glucose-limited environment. Previously, we reported that inhibition of PFK-1 by Snail increased metabolic flux into the pentose phosphate pathway (PPP) for NADPH generation. Interestingly, suppression of FBP-1 followed by decreased endoergonic gluconeogenesis can contribute to activating nonoxidative PPP by bi-directional transketolase and transaldolase (Berg JM, Tymoczko JL, & Stryer L. Biochemistry, 7th ed. 2010, p606-p609). We have recently summarized those catabolic circuits, as referenced in this manuscript (Cho ES et al, Biomol Ther, 2018, 26, 29-38). The suppression of PFK-1 and FBP-1 may utilize Mode3 and Mode4 circuits, respectively (please see Fig. 2 in Cho ES et al). These suggest the possibility of several catabolic circuits to provide ATP and NADPH according to cancer subtypes or oncogenic activation. To avoid confusing readers, we have added detailed comments on metabolic outcomes by inhibition of PFK-1 or FBP-1 in the Discussion section.

- The authors use references in many instances all throughout the paper that do not fit with what is stated in the text.

Authors' Response) We have thoroughly and carefully reviewed the manuscript again. Please specify the reference if further correction is required.

- END -

May 18, 2020

RE: Life Science Alliance Manuscript #LSA-2020-00683-TR

Prof. Jong In Yook
Yonsei University College of Dentistry
Department of Oral Pathology
50-1 Yonsei-ro, Seodaemoon-gu
Seoul 03722
Korea (South), Republic of

Dear Dr. Yook,

Thank you for submitting your revised manuscript entitled "Snail augments fatty acid oxidation by suppression of mitochondrial ACC2 during cancer progression". We have re-evaluated your work within our editorial team. While we think that the findings are still difficult to reconcile with the previous literature, we appreciate your response to this concern and would be happy to publish your paper in Life Science Alliance pending final revisions necessary to meet our formatting guidelines:

- LC-MS raw data should get included or deposited in a repository (eg. MetaboLights).
- Since data on the Broad GDAC firehose changes with time, it is important to include the raw data used in your analyses to allow others to reproduce your findings. Please do so by either providing the raw files as "data set" files or by depositing them in a repository and provide an accession code in the manuscript text.
- Please list in each figure legend the statistical test used and p-values (description is lacking in many places). It seems that those currently already described, are not necessarily adequate ones - please revise.
- All figures need to get uploaded as individual files, including the supplementary files; the legends (including supplementary figure legends) should remain in the main manuscript docx file; please use the same style for the legends in both main and suppl figure legends
- We can only publish figures that adhere to our guidelines, please revise accordingly (figure 1 spans two pages at the moment, figure quality not sufficient (blurry) in some instances)
- Please add a scale bar to Fig S1H and increase visibility of the scale bar in Fig 6E

A. FINAL FILES:

B. MANUSCRIPT ORGANIZATION AND FORMATTING:

Sincerely,

May 22, 2020

RE: Life Science Alliance Manuscript #LSA-2020-00683-TRR

Prof. Jong In Yook
Yonsei University College of Dentistry
Department of Oral Pathology
50-1 Yonsei-ro, Seodaemun-gu
Seoul 03722
Korea (South), Republic of

Dear Dr. Yook,

Thank you for submitting your Research Article entitled "Snail augments fatty acid oxidation by suppression of mitochondrial ACC2 during cancer progression". I appreciate the introduced changes and it is a pleasure to let you know that your manuscript is now accepted for publication in Life Science Alliance. Congratulations on this interesting work.

DISTRIBUTION OF MATERIALS:

Again, congratulations on a very nice paper. I hope you found the review process to be constructive and are pleased with how the manuscript was handled editorially. We look forward to future exciting submissions from your lab.

Sincerely,

Reilly Lorenz
Editorial Office Life Science Alliance
Meyerhofstr. 1
69117 Heidelberg, Germany
t +49 6221 8891 414
e contact@life-science-alliance.org
www.life-science-alliance.org